# ALT-MAS: A Data-Efficient Framework for Active Testing of Machine Learning Algorithms

## Abstract

Machine learning models are being used extensively in many important areas, but there is no guarantee that a model will always perform well or as its developers intended. Understanding the correctness of a model is crucial to prevent potential failures that may have significant detrimental impact in critical application areas. In this paper, we propose a novel framework to efficiently test a machine learning model using only a small amount of labelled test data. The core idea is to efficiently estimate the metrics of interest for a model-under-test using Bayesian neural network. We develop a novel methodology to incorporate the information from the model-under test into the Bayesian neural network training process. We also devise an entropy-based sampling strategy to sample the data point such that the proposed framework can give accurate estimations for the metrics of interest. Finally, we conduct an extensive set of experiments to test various machine learning models for different types of metrics. Our experiments with multiple datasets show that given a testing budget, the estimation of the metrics by our method is significantly better compared to existing state-of-the-art approaches.

## 1 Introduction

Today, supervised machine learning models are employed across sectors to assist humans in making important decisions. Understanding the correctness of a model is thus crucial to avoid potential (and severe) failures. In practice, however, it is not always possible to accurately evaluate the model's correctness using the held-out training data in the development process (Sawade et al., 2010). Consider a hospital that buys an automated medical image classification system. The supplier will provide a performance assessment, but this evaluation may not hold in this new setting as the supplier and the hospital data distributions may differ. Similarly, an enterprise that develops a business prediction system might find that the performance changes significantly over time as the input distribution shifts from the original training data. In these cases, the model performance needs to be re-evaluated as the assessments provided from the supplier or from the development process can be inaccurate. To accurately evaluate the model performance, new labelled data points from the deployment area are needed. But the process of labelling is expensive as one would usually need a large number of test instances. *Thus the open question is how to test the performance of a machine learning model (model-under-test) with parsimonious use of labelled data from the deployment area.*

This work focuses on addressing this challenge treating the model-under-test as a black-box as in common practice one only has access to the model outputs. One previous approach aims to estimate a risk score which is a function of the model-under-test output and the ground-truth (akin to metric) using limited labelled data (Sawade et al., 2010). However, the approach has only been shown to be tractable for some specific risk functions (e.g. accuracy). Another approach in (Gopakumar et al., 2018) suggested to search for the worst case model performance using limited labelled data, however, we posit that using worst case to assess the goodness of a model-under-test is an overkill because the worst case is often just an outlier. Recently, (Schelter et al., 2020) learns to validate the model without labelled data by generating a synthetic dataset representative of the deployment data. The restrictive assumption is that it requires domain experts to provide a set of data generators, a task usually infeasible in reality.

We propose a *scalable data-efficient framework* that can assess the performance of a black-box model-under-test on any metric (that is applicable for black-box models) without prior knowledge from users. Furthermore, our framework can estimate multiple metrics simultaneously. The motivation for evaluating one or multiple metrics is inspired by the current practice of users who need to assess the model-under-test on one or varied aspects that are important to them. For instance, for a classification system, the user might want to solely check the overall accuracy or simultaneously check the overall accuracy, macro-precision (recall) and/or the accuracies of some classes of interest.

To achieve sample efficiency, we formulate our testing framework as an active learning (AL) problem (Cohn et al., 1996). First, a small subset of the test dataset is labelled, and a surrogate model is learned from this subset to predict the ground truth of the unlabelled data points in the test dataset. Second, an acquisition function is constructed to decide which data point in the test dataset should be chosen for labelling. The data point selected by the acquisition is sent to an external oracle for labelling, and is then added to the labelled set. The process is conducted iteratively until the labelling budget is depleted. The metrics of interest are then estimated using the learned surrogate model.

With this framework, one choice is to use a standard AL method to learn a surrogate model that accurately predicts the labels of all the data points in the test dataset, however, this choice is not optimal. To efficiently estimate the metrics of interest, the surrogate model should not need to accurately predict the labels of all the data points; it only needs to accurately predict the labels of those data points that contribute significantly to the accuracy of the metric estimations. For our active testing framework, we first propose a method to train the surrogate model that can provide high metric estimation accuracy (using limited number of labelled data) by incorporating information from the model-under-test. Second, we derive an entropy-based acquisition function that can select the data points for whom labels should be acquired so as to enable maximal reduction in the estimation uncertainty of the metric of interest. We then use this computed entropy to generalize our framework to be able to work with multiple metrics. Finally, we demonstrate the efficacy of our proposed testing framework using various models-under-test and a wide range of metric sets on different datasets. In summary, our main contributions are:

1. ALT-MAS, a data-efficient testing framework that can accurately estimate the performance of a machine learning model;

2. A novel approach to train the BNN so as to accurately estimate the metrics of interest;

3. A novel sampling methodology so as to estimate the metrics of interest efficiently; and,

4. Demonstration of the empirical effectiveness of our proposed machine learning testing framework on various models-under-test for a wide range of metrics and different datasets.

## 2 PROBLEM FORMULATION AND BACKGROUND

### 2.1 PROBLEM FORMULATION

Let us assume we are given a black-box model-under-test $\mathcal{A}$ that gives the prediction $\mathcal{A}(x)$ for an input $x$, with $\mathcal{A}(x) \in \mathcal{C} = \{1, \ldots, C\}$. Let us also assume we have access to (i) an unlabelled test dataset $\mathcal{X} = \{x_i\}_{i=1}^{N}$, and, (ii) an oracle that can provide the label $y_x$ for each input $x$ in $\mathcal{X}$. Given a set of performance metrics $\{Q_k\}_{k=1}^{K}$, $Q_k : \mathbb{R}^N \times \mathbb{R}^N \to \mathbb{R}$, the goal is to *efficiently* estimate the values of these metrics, $\{Q_k^*\}_{k=1}^{K}$, when evaluating the model-under-test $\mathcal{A}$ on the test dataset $\mathcal{X}$. That is, we aim to estimate,

$$Q_k^* = Q_k(\mathcal{A}_\mathcal{X}, \mathcal{Y}_\mathcal{X}), \quad k = 1, \ldots, K, \tag{1}$$

with $\mathcal{A}_\mathcal{X} = \{\mathcal{A}(x)\}_{x \in \mathcal{X}}$ and $\mathcal{Y}_\mathcal{X} = \{y_x\}_{x \in \mathcal{X}}$, using the minimal number of oracle queries.

In this work, we focus on classifiers because they are common supervised learning models and also the target models of most machine learning testing papers (Zhang et al., 2019). Besides, it is also worth noting that, as we only have access to the outputs of the black-box classifier, the metrics $\{Q_k\}$ must be those that can be computed using solely the classifier outputs $\mathcal{A}_\mathcal{X}$ and the ground-truth labels $\mathcal{Y}_\mathcal{X}$ of the test dataset $\mathcal{X}$. Examples of $Q_k$ include the accuracy, error rate, per-class precision/recall, macro precision/recall, $F_\beta$ score, etc. This is to distinguish with the metrics that require information from the classifier internal structure such as the log-loss metric.

## 2.2 BACKGROUND

Bayesian neural networks (BNNs) are special neural networks that maintain a distribution over its parameters (MacKay, 1992; Neal, 1995). Specifically, given the training data $\mathcal{D}_{tr} = \{x_i, y_i\}_{i=1}^N$, a BNN can provide the posterior distribution $p^*(\omega|\mathcal{D}_{tr})$ with $\omega$ being the neural network weights. In practice, performing exact inference to obtain $p^*(\omega|\mathcal{D}_{tr})$ is generally intractable, hence we use a variational approximation technique to approximate this posterior. In particular, we employ the MC-dropout method (Gal & Ghahramani, 2016) as it is known to be both scalable and theoretically guaranteed in terms of inferring the true model posterior distribution $p^*(\omega|\mathcal{D}_{tr})$. That is, the MC-dropout method is equivalent to performing approximate variational inference to find a distribution in a tractable family that minimizes the Kullback-Leibler divergence to the true model posterior.

## 3 ACTIVE TESTING WITH METRIC-AWARE SAMPLING STRATEGY

Our active testing framework is summarized as follows. First, a small subset $\mathcal{X}_l$ of the test dataset $\mathcal{X}$ is labelled to construct a labelled set $\mathcal{D}_l = \{\mathcal{X}_l, \mathcal{Y}_{\mathcal{X}_l}\}$, where $\mathcal{Y}_{\mathcal{X}_l}$ denotes the labels provided by the oracle for $\mathcal{X}_l$; and a BNN $\mathcal{B}_\omega$ (with parameter $\omega$) is learned from $\mathcal{D}_l$ to predict the labels of the unlabelled data points in $\mathcal{X}$. Second, an acquisition function is constructed based on the BNN $\mathcal{B}_\omega$, the characteristics of the metric set, and the model-under-test outputs $\mathcal{A}_\mathcal{X}$ to decide which data point is to be labelled so as to maximally reduce the uncertainty in the metric estimations. This data point is sent for labelling, and is added to the labelled set $\mathcal{D}_l$. This process is conducted iteratively until the labelling budget is depleted. The metrics of interest are estimated using the BNN $\mathcal{B}_\omega$.

In this section, we propose a method to train a BNN that can give accurate metric estimations from a limited number of labelled data (**Section 3.1**), a method to estimate the metrics of interest given the BNN (**Section 3.2**), and a method to sample the most informative data point to maximize the estimation accuracy of a specific metric (**Section 3.3**) or a set of metrics (**Section 3.4**).

### 3.1 BAYESIAN NEURAL NETWORK TRAINING METHODOLOGY

Given the labelled set $\mathcal{D}_l$ and the model-under-test outputs $\mathcal{A}_\mathcal{X}$, the goal is to train a BNN $\mathcal{B}_\omega$ such that the corresponding metric estimations are most accurate. Training BNN using solely the labelled set $\mathcal{D}_l$ might not result in accurate enough metric estimations. Thus, to improve the metric estimation accuracy, we propose to incorporate the information from the model-under-test outputs $\mathcal{A}_\mathcal{X}$ into the BNN training process. In particular, using the labelled set $\mathcal{D}_l$, we also train a binary classifier $\mathcal{C}_\eta$ that aims to predict the data points in the test dataset for which the model-under-test agrees with the ground-truth. Using the predictions by the classifier $\mathcal{C}_\eta$, we then construct an *augmented labelled set* $\mathcal{S}_l = \{\mathcal{X}_S, \mathcal{Y}_{\mathcal{X}_S}\}$ where $\mathcal{X}_S$ are all the data points in the test dataset $\mathcal{X}$ that $\mathcal{C}_\eta$ identifies the model-under-test predictions are accurate, and $\mathcal{Y}_{\mathcal{X}_S}$ are the corresponding model-under-test outputs of $\mathcal{X}_S$. The BNN is then trained using both the labelled set $\mathcal{D}_l$ and the augmented labelled set $\mathcal{S}_l$.

To train the binary classifier $\mathcal{C}_\eta$, we first split the labelled set $\mathcal{D}_l$ into two parts: training and validation, and then train $\mathcal{C}_\eta$ on the training part whilst tuning the softmax probability threshold using the validation part so that $\mathcal{C}_\eta$ achieves *the highest precision* on the validation part. This is because we want $\mathcal{C}_\eta$ to choose a data point only when it is most certain that the ground-truth and the model-under-test output of that data point is the same. Besides, as the precision of $\mathcal{C}_\eta$ is rarely $100\%$, thus, after obtaining the set of data points provided by $\mathcal{C}_\eta$, we only take $N_s$ data points from this set with the highest softmax probability. The number $N_s$ is computed by multiplying the precision of $\mathcal{C}_\eta$ on the validation part with the cardinality of the original predicted set. For example, if the precision of the classifier $\mathcal{C}_\eta$ is $50\%$ on the validation part and the original predicted set consisting of $100$ data points, the final augmented labelled set $\mathcal{S}_l$ only consists of $50$ data points with the highest softmax probability. Finally, the binary classification problem can be imbalanced, particularly when the model-under-test is very accurate or very bad. Hence, when training $\mathcal{C}_\eta$, we employ the over-sampling technique (for the minority class) to ensure the training data of the binary classification problem to be balanced, i.e. the cardinalities of the majority and minority classes are equal.

**Remark 3.1.** With this training methodology, the more accurate the model-under-test, the more accurate the BNN $\mathcal{B}_\omega$. In case when the model-under-test is bad, the augmented labelled set $\mathcal{S}_l$ does not consist of many elements, thus, the BNN accuracy does not improve much compared to when training solely using the labelled set $\mathcal{D}_l$. However, in this case, the BNN does not need to have high

accuracy in order to accurately estimate the metrics. Specifically, for any data point for which the model-under-test disagrees with the ground truth, the BNN does not need to accurately predict its label. That is, even when the BNN predicts other labels (except the model-under-test output label), the metric estimation is still accurate (more detailed examples in Section C.3 of the appendix).

**Remark 3.2.** For simplicity, we suggest to set the architecture of the binary classifier and BNN to be same. For example, if the BNN is a 2-layer MLP, then the binary classifier is also a 2-layer MLP.

## 3.2 Metric Estimation with Bayesian Neural Network

For a metric $Q_k$, given the model-under-test outputs $\mathcal{A}_\mathcal{X}$ and the labelled set $\mathcal{D}_l$,[1] the true value $Q_k^*$ of this metric generally cannot be computed because $\mathcal{Y}_{\mathcal{X}_{ul}}$ (the corresponding labels of the unlabelled set $\mathcal{X}_{ul}$) is unknown. However, if the true posterior distribution $p^*(\omega|\mathcal{D}_l)$ of the BNN is known, then $Q_k^*$ can be computed as,

$$Q_k^* = Q_k(\mathcal{A}_\mathcal{X}, \mathcal{Y}_\mathcal{X}) = Q_k(\mathcal{A}_\mathcal{X}, [\mathcal{Y}_{\mathcal{X}_l}, \mathcal{Y}_{\mathcal{X}_{ul}}]) = \mathbb{E}_{\hat{\mathcal{Y}}_{\mathcal{X}_{ul}} \sim p(\hat{y}_{\mathcal{X}_{ul}}|\mathcal{X}_{ul})}[Q_k(\mathcal{A}_\mathcal{X}, [\mathcal{Y}_{\mathcal{X}_l}, \hat{\mathcal{Y}}_{\mathcal{X}_{ul}}])]$$
$$= \int Q_k(\mathcal{A}_\mathcal{X}, [\mathcal{Y}_{\mathcal{X}_l}, \hat{\mathcal{Y}}_{\mathcal{X}_{ul},\omega}])p^*(\omega|\mathcal{D}_l)\,\mathrm{d}\omega, \tag{2}$$

where $\hat{\mathcal{Y}}_{\mathcal{X}_{ul}}$ is a random variable representing the possible labels of $\mathcal{X}_{ul}$, $\hat{\mathcal{Y}}_{\mathcal{X}_{ul},\omega}$ denotes the predicted labels of $\mathcal{X}_{ul}$ using the BNN weights $\omega$, i.e. $\hat{\mathcal{Y}}_{\mathcal{X}_{ul},\omega} = \{\hat{y}_{x,\omega}\}_{x \in \mathcal{X}_{ul}}$, and $\hat{y}_{x,\omega} = \arg\max_{c=1,\dots,C} p(y_x = c|x,\omega), \forall x \in \mathcal{X}_{ul}$. Substitute $p^*(\omega|\mathcal{D}_l)$ by the MC-dropout variational distribution $q_\theta(\omega|\mathcal{D}_l)$ (Gal & Ghahramani, 2016), we can approximate $Q_k^*$ as,

$$\hat{Q}_k = \int Q_k(\mathcal{A}_\mathcal{X}, [\mathcal{Y}_{\mathcal{X}_l}, \hat{\mathcal{Y}}_{\mathcal{X}_{ul},\omega}])q_\theta(\omega|\mathcal{D}_l)\,\mathrm{d}\omega \approx \frac{1}{M}\sum_{j=1}^M Q_k(\mathcal{A}_\mathcal{X}, [\mathcal{Y}_{\mathcal{X}_l}, \hat{\mathcal{Y}}_{\mathcal{X}_{ul},\hat{\omega}_j}]), \tag{3}$$

where $\{\hat{\omega}_j\}_{j=1}^M$ are $M$ stochastic forward passes from the distribution $q_\theta(\omega)$.

**Remark 3.3.** It is worth noting that the value of metric $Q_k$ can also be estimated by computing $\mathbb{E}[Q_k(\mathcal{A}_\mathcal{X}, \hat{\mathcal{Y}}_{\mathcal{X},\omega})]$ where $\hat{\mathcal{Y}}_{\mathcal{X},\omega}$ are the predictions of the whole test dataset $\mathcal{X}$ using $\omega$. However, in this case, the estimation might be biased, i.e. the estimation might not converge to the true value of the metric when the labelled set is the whole test dataset. In fact, this estimation depends on the quality of the $M$ stochastic forward passes $\{\hat{\omega}_j\}$. In contrast, with the approximation proposed in Eqs. (2) and (3), the estimation becomes increasingly less biased and when all the data points in the test dataset is labelled, the estimation obtained by our method is equal to the true metric value.

## 3.3 Sampling Methodology for a Single Metric

In the sequel, we use $\mathcal{D}_l^t = \{\mathcal{X}_l^t, \mathcal{Y}_{\mathcal{X}_l^t}\}$ and $\mathcal{X}_{ul}^t = \mathcal{X} \setminus \mathcal{X}_l^t$ to denote the labelled and unlabelled set obtained after iteration $t$, respectively. We also use $\mathcal{B}_\omega^t$ to denote the BNN trained on $\mathcal{D}_l^t$ and $\mathcal{Y}_{\mathcal{X}_{ul}^t}$ to denote the labels corresponding to the unlabelled set $\mathcal{X}_{ul}^t$. For a metric $Q_k$, at iteration $t + 1$, given the model-under-test outputs $\mathcal{A}_\mathcal{X}$ and the labelled set $\mathcal{D}_l^t$, the goal of the sampling process is to select a data point $x_t^*$ to label so as to maximally increase the metric estimation accuracy. This can be considered equivalent to sampling the data point $x_t^*$ such that knowing its label results in maximal uncertainty reduction for the metric estimation. To identify $x_t^*$, our key idea is to (i) evaluate how much the uncertainty of each data point's label contributes to the uncertainty of the metric estimation, and, (ii) sample the data point causing the highest uncertainty in the metric estimation.

**To solve (i)**, we derive a formulation that captures the relation between the uncertainty of a data point's label and the uncertainty of the metric value. We construct this formulation based on Eq. (2), but we set the data point of interest (e.g. $x$) as a variable while fixing other data points (e.g. $\mathcal{X}_{ul}^t \setminus x$) as their expected values. Specifically, let us denote $\hat{y}_x$ as the random variable representing all the possible labels of a data point $x$, then we define a new random variable $\tilde{Q}_k(x)$ as follows,

$$\tilde{Q}_k(x) = \mathbb{E}_{\hat{\mathcal{Y}}_{\mathcal{X}_{ul}^t \setminus x} \sim p(\hat{y}_{\mathcal{X}_{ul}^t \setminus x}|\mathcal{X}_{ul}^t \setminus x, \mathcal{D}_l^t)}[Q_k(\mathcal{A}_\mathcal{X}, [\mathcal{Y}_{\mathcal{X}_l^t}, \hat{\mathcal{Y}}_{\mathcal{X}_{ul}^t \setminus x}, \hat{y}_x])], \quad \forall x \in \mathcal{X}_{ul}^t, \tag{4}$$

where $\hat{\mathcal{Y}}_{\mathcal{X}_{ul}^t \setminus x}$ denotes the random variable representing the possible labels of $\mathcal{X}_{ul}^t \setminus x$ conditioned on $\mathcal{D}_l^t$. This random variable represents the possible values of metric $Q_k$ for each data point $x$.

---

[1] $\mathcal{D}_l$ includes the augmented data set generated using the proposed method in Secion 3.2

**To solve (ii)**, we formulate an acquisition function that can suggest which data point that causes the highest uncertainty in the metric estimation. We define it as the mutual information (Houlsby et al., 2011; Gal et al., 2017) between $\tilde{Q}_k(x)$ and the BNN parameters $\omega$, i.e.,

$$\mathbb{I}[\tilde{Q}_k(x); \omega | x, \mathcal{D}_l^t] = \mathbb{H}[\tilde{Q}_k(x) | x, \mathcal{D}_l^t] - \mathbb{E}_{\omega \sim p(\omega | \mathcal{D}_l^t)}[\mathbb{H}[\tilde{Q}_k(x) | x, \omega]]. \tag{5}$$

We use the mutual information as it is one of the most common criteria in deep Bayesian active learning to represent uncertainty (Gal et al., 2017). The first term of the acquisition function in Eq. (5) is the entropy of the metric estimation and the second term is an expectation of the entropy of the metric estimation over the posterior of the model parameters. The data point that maximizes $\mathbb{I}[\tilde{Q}_k(x); \omega | x, \mathcal{D}_l^t]$ is the data point for which the model has many possible values for metric estimation, i.e. the posterior draws have disagreement.

We can approximate $\mathbb{I}[\tilde{Q}_k(x); \omega | x, \mathcal{D}_l^t]$ by using the posterior MC-dropout distribution $q_\theta(\omega | \mathcal{D}_l^t)$. First, we show how to compute the first term on the right hand side of Eq. (5). Since for each data point $x$, $\hat{y}_x$ is a discrete random variable with $C$ distinct values, so $\tilde{Q}_k(x)$ is also a discrete random variable with at most $C$ distinct values. Therefore, its entropy $\mathbb{H}[\tilde{Q}_k(x) | x, \mathcal{D}_l^t]$ can be computed as,

$$\mathbb{H}[\tilde{Q}_k(x) | x, \mathcal{D}_l^t] = -\sum_{q \in \mathcal{Q}} p(\tilde{Q}_k(x) = q | x, \mathcal{D}_l^t) \log p(\tilde{Q}_k(x) = q | x, \mathcal{D}_l^t), \tag{6}$$

where $\mathcal{Q}$ consists of all the possible values of $\tilde{Q}_k(x)$ when $\hat{y}_x \in \{1, ..., C\}$. By using the union bound, $p(\tilde{Q}_k(x) = q | x, \mathcal{D}_l^t)$ can be expressed as $\sum_{h \in \tilde{Q}_k^{-1}(q)} p(\hat{y}_x = h | x, \mathcal{D}_l^t)$ where $\tilde{Q}_k^{-1}(q)$ is the inverse function that maps the value of $\tilde{Q}_k(x)$ to $\hat{y}_x$. Given $M$ stochastic forward passes $\{\hat{\omega}_j\}_{j=1}^M$ from $q_\theta(\omega | \mathcal{D}_l^t)$, $\mathbb{H}[\tilde{Q}_k(x) | x, \mathcal{D}_l^t]$ can finally be approximated as,

$$\begin{aligned}\mathbb{H}[\tilde{Q}_k(x) | x, \mathcal{D}_l^t] \approx -\sum_{q \in \mathcal{Q}} &\left( \left( \sum_{h \in \tilde{Q}_k^{-1}(q)} (\sum_{j=1}^M p(\hat{y}_x = h | x, \hat{\omega}_j))/M \right) \right. \\ &\left. \times \log \left( \sum_{h \in \tilde{Q}_k^{-1}(q)} (\sum_{j=1}^M p(\hat{y}_x = h | x, \hat{\omega}_j))/M \right) \right),\end{aligned} \tag{7}$$

where $\tilde{Q}_k(x)$ can be approximated as $\tilde{Q}_k(x) \approx \mathbb{E}_{\omega \sim q_\theta(\omega | \mathcal{D}_l^t)} [Q_k(\mathcal{A}_\mathcal{X}, [\mathcal{Y}_{\mathcal{X}_l^t}, \hat{\mathcal{Y}}_{\mathcal{X}_{ul}^t \backslash x, \omega}, \hat{y}_x])] \approx (\sum_{j=1}^M Q_k(\mathcal{Y}_\mathcal{A}, [\mathcal{Y}_l^t, \hat{\mathcal{Y}}_{\mathcal{X}_{ul}^t \backslash x, \hat{\omega}_j}, \hat{y}_x]))/M$, with $\hat{\mathcal{Y}}_{\mathcal{X}_{ult}^t \backslash x, \hat{\omega}_j}$ denoting the predicted labels for $\mathcal{X}_{ul}^t \backslash x$ given the parameter $\hat{\omega}_j$. The second term, $\mathbb{E}_{\omega \sim p(\omega | \mathcal{D}_l^t)}[\mathbb{H}[\tilde{Q}_k(x) | x, \omega]]$, can also be approximated in a similar fashion (detailed formula is in Section B.2 of the appendix).

Finally, the data point to be sampled is the one with the highest value of $\mathbb{I}[\tilde{Q}_k(x); \omega | x, \mathcal{D}_l^t]$, i.e.,

$$x_{t+1}^* = \text{argmax}_{x \in \mathcal{X}_{ul}^t} \mathbb{I}[\tilde{Q}_k(x); \omega | x, \mathcal{D}_l^t]. \tag{8}$$

**Remark 3.4.** In the case $\text{argmax}_{x \in \mathcal{X}_{ul}^t} \mathbb{I}[\tilde{Q}_k(x); \omega | x, \mathcal{D}_l^t]$ contains multiple elements, we select the data point that has the highest classification mutual information $\mathbb{I}[\hat{y}_x; \omega | x, \mathcal{D}_l^t]$, where $\mathbb{I}[\hat{y}_x; \omega | x, \mathcal{D}_l^t]$ is the standard mutual information for classification problems (BALD) (Gal et al., 2017).

### 3.4 SAMPLING METHODOLOGY FOR MULTIPLE METRICS

Given a set of multiple metrics $\{Q_k\}_{k=1}^K$, at iteration $t + 1$, the goal is to sample the data point to maximally increase the estimation accuracy of all the metrics. As proposed in Section 3.3, for each metric $Q_k$, we can sample the data point with the highest value of $\mathbb{I}[\tilde{Q}_k(x); \omega | x, \mathcal{D}_l^t]$. Extending this idea to multiple metrics, we can sample the data point with the highest sum of $\mathbb{I}[\tilde{Q}_k(x); \omega | x, \mathcal{D}_l^t]$,

$$x_{t+1}^* = \text{argmax}_{x \in \mathcal{X}_{ul}^t} \sum_{k=1}^K \mathbb{I}[\tilde{Q}_k(x); \omega | x, \mathcal{D}_l^t], \tag{9}$$

where $\mathbb{I}[\tilde{Q}_k(x); \omega | x, \mathcal{D}_l^t]$ defined as in Eq. (5).

Our proposed framework **A**ctive **L**earning for **T**esting with **M**etric-**A**ware **S**ampling strategy, **ALT-MAS**, is summarized in Algorithm 1.

---

**Algorithm 1** The Proposed **ALT-MAS** Algorithm

---

1: **Input:** Model-under-test $\mathcal{A}$, test dataset $\mathcal{X}$, initial labelled set $\mathcal{D}_l^0 = \{\mathcal{X}_l^0, \mathcal{Y}_l^0\}$ ($\mathcal{X}_l^0 \in \mathcal{X}$), the Bayesian neural network $\mathcal{B}_\omega$, labelling budget $T$, a set of metrics of interest $\{Q_k\}_{k=1}^K$

2: **Output:** The estimations of the metrics in $\{Q_k\}_{k=1}^K$

3: Initialize $\mathcal{X}_{ul}^0 = \mathcal{X} \setminus \mathcal{X}_l^0$.

4: **for** $t = 1, 2, \ldots, T$ **do**

5:     /* Generate stochastic forward passes $\hat{\omega}_j \sim q_\theta(\omega | \mathcal{D}_l^{t-1})$ */

6:     Train the binary classifier $\mathcal{C}_\eta^{t-1}$ using $\mathcal{D}_l^{t-1}$

7:     Generate the augmented labelled set $\mathcal{S}_l^{t-1}$ using $\mathcal{C}_\eta^{t-1}$

8:     Train the Bayesian neural network $\mathcal{B}_\omega^{t-1}$ from $\mathcal{D}_l^{t-1}$ and $\mathcal{S}_l^{t-1}$

9:     Generate $M$ stochastic forward passes $\{\hat{\omega}_j\}_{j=1}^M$ from $\mathcal{B}_\omega^{t-1}$ using MC-dropout

10:    /* Compute the distribution of random variables $\tilde{Q}_k(x)$ for every $x$ */

11:    **for** $k = 1, 2, \ldots, K$ **do**

12:       For each $x \in \mathcal{X}_{ul}^{t-1}$, compute $\tilde{Q}_k(x)$ with all $\hat{y}_x \in \{1, ..., C\}$ using Eq. (4) and $\{\hat{\omega}_j\}_{j=1}^M$

13:       For each $x \in \mathcal{X}_{ul}^{t-1}$, compute the set $\mathcal{Q}_k(x)$, that consists of all possible values of $\tilde{Q}_k(x)$

14:    **end for**

15:    /* Find the data point that maximizes the mutual information function */

16:    Solve $x_t^* = \mathrm{argmax}_{x \in \mathcal{X}_{ul}^{t-1}} \sum_{k=1}^K \mathbb{I}[\tilde{Q}_k(x); \omega | x, \mathcal{D}_l^{t-1}]$ as formulated in Eq. (5)

17:    Ask the oracle for a label of $x_t^*$

18:    Update $\mathcal{D}_l^t = \mathcal{D}_l^{t-1} \cup \{x_t^*, y_{x_t^*}\}$ and $\mathcal{X}_{ul}^t = \mathcal{X}_{ul}^{t-1} \setminus x_t^*$

19: **end for**

20: Compute the metric estimations $\{\hat{Q}_k\}_{k=1}^K$ using Eq. (3)

---

### 3.5 THEORETICAL TIME COMPLEXITY OF THE PROPOSED ACQUISITION FUNCTIONS

Let us denote $\mathcal{O}(a_k | \mathcal{X}|)$ as the time complexity when computing metric $Q_k$ on test dataset $\mathcal{X}$ (e.g. when $Q_k$ is the accuracy metric, $a_k$ is 1). Then the time complexity of ALT-MAS (Eq. 8) is $\mathcal{O}(C^2 |\mathcal{X}| + (a_k + 2)MC|\mathcal{X}|)$, where $C$ is the number of classes of the model-under-test, $M$ is the number of Monte Carlo forward passes, and $|\mathcal{X}|$ is the cardinality of $\mathcal{X}$. This time complexity is computed by splitting the computation of ALT-MAS into 3 steps: (1) computing $\tilde{Q}_k(x) \; \forall x \in \mathcal{X}$ when $\hat{y}_x \in \{1, ., C\}$ - time complexity $\mathcal{O}(\alpha_k MC|\mathcal{X}|)$, (2) computing probability distribution of $\tilde{Q}_k(x) \; \forall x \in \mathcal{X}$ - time complexity $\mathcal{O}(C^2 |\mathcal{X}| + MC|\mathcal{X}|)$, and, (3) computing the entropy - time complexity $\mathcal{O}(MC|\mathcal{X}|)$. For a set of $K$ metrics, the time complexity of ALT-MAS (Eq. 9) is $\mathcal{O}(KC^2 |\mathcal{X}| + 2KMC|\mathcal{X}| + \sum_{k=1}^K a_k MC|\mathcal{X}|)$, i.e. it is linear in the number of instances in $\mathcal{X}$.

## 4 EXPERIMENTAL RESULTS

We evaluate our active testing framework using various models-under-test and metric sets on the datasets MNIST and CIFAR10. Our experiments aim to answer:

1. *Does our active testing framework perform better than traditional machine learning testing approaches?* This checks whether a testing framework based on AL is necessary.
2. *Does our active testing framework perform better than existing AL approaches when estimating metric values?* This is to show that a suitable BNN training methodology and a sampling approach that reduces the uncertainty of the metric estimation are needed.
3. *Does our proposed sampling method work well for different sets of metrics and models-under-test?* This is to ensure robustness across scenarios.
4. *How is the performance of the BNNs?* This checks whether a complex BNN is needed for our active testing framework to be successful.

To answer questions (1) and (2), we compare our method **ALT-MAS** with two baselines: (i) **Tradition**: the traditional method where the metrics are computed using their mathematical formula with all the labelled data up to the current iteration, and the labelled data is picked randomly from the whole test dataset; (ii) **BALD**: the state-of-the-art deep Bayesian AL method (Gal et al., 2017), the metrics are estimated using the predicted label of the BNN trained with BALD. Note that here, we

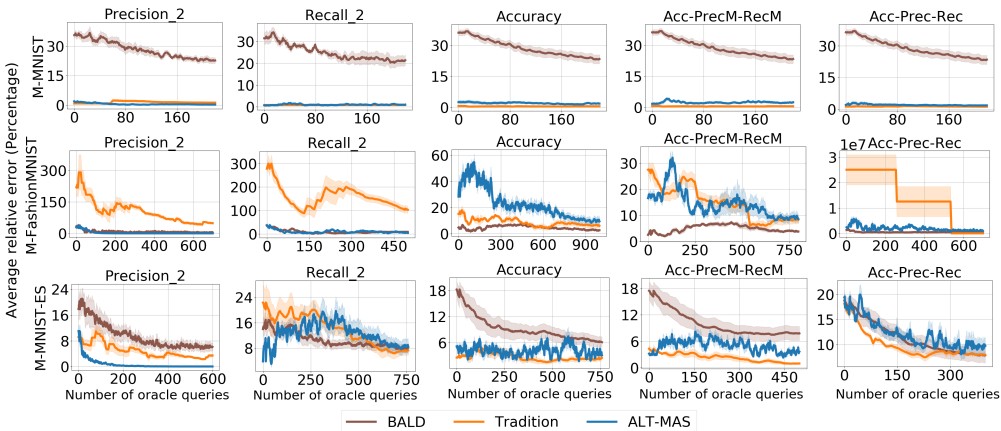

Figure 1: Average relative error of the estimated metrics, for each combination of model-under-test (*M-MNIST*, *M-FashionMNIST*, & *M-MNIST-ES*) and metric set. Curves are smoothed over a sliding window of 10 oracle queries (iterations). Plotting mean and standard error over 3 repetitions. Better methods are those that converge faster to zero (Best seen in color).

don't compare with the traditional AL methods as it has been shown that deep Bayesian AL methods outperform traditional AL methods (Settles, 2010; Gal et al., 2017).

To answer questions (3) and (4), for each dataset, we evaluate our proposed machine learning testing framework using various types of models-under-test and metric sets. Since our testing framework depends solely on the model-under-test outputs, we evaluate our proposed framework with different models-under-test that have different levels of accuracy (e.g. good, average, and bad). This is to simulate the reality when the user receives a black-box model-under-test and they want to know if that model-under-test is good or bad using their criteria (i.e. metrics). The metric sets consist of either a single metric or multiple metrics, and the metrics are of different types: per-class metrics and overall metrics. For each combination of dataset, model-under-test and metric set, we repeat the experiment 3 times. Finally, we investigate the prediction performance of the BNNs used in all these experiments. All the experiments are running on multiple servers where each server has multiple Tesla V100 SXM2 32GB GPUs. All the source codes are implemented in Tensorflow 1.15.0 and will be publicly available after the acceptance of this paper.

## 4.1 MNIST DATASET

We aim to estimate the performance of various models-under-test using different metric sets on the MNIST test dataset (LeCun et al., 2010). Our experiment setup is as follows. Firstly, we use a standard convolutional neural network (CNN),[2] and then train it on the train MNIST dataset to construct a good model-under-test (*M-MNIST*). Secondly, we either reduce the number of training epochs, retrain the CNN to construct average models-under-test (*M-MNIST-ES*). Thirdly, we train the CNN on the train FashionMNIST dataset (Xiao et al., 2017) (completely different from MNIST) to generate a bad model-under-test (*M-FashionMNIST*). Details of these models are in Section C.1 of the appendix. For each model-under-test, we aim to estimate various sets of common metrics: i) 3 metric sets containing solely one metric in each set: precision of class 2, recall of class 2,[3] and overall accuracy, and, ii) 2 metric sets consisting of multiple metrics: a set of 3 metrics (accuracy, macro-precision, macro-recall), and a set of 21 metrics (accuracy, precision and recall of each class). The BNN and binary classifier are mutilayer perceptrons (MLP) with 2 layers and dropout in each layer (details are in Section C.1 of the appendix).

In Figure 1, we report the average relative errors of the estimated metrics for each combination of model-under-test and metric set. First, we can see that the method **Tradition** is not consistent.

---

[2]The model is implemented in tensorflow but use the official network architecture published on keras.com for MNIST dataset, as there is no official source code on tensorlow.org for MNIST dataset.

[3]Class 2 is a random choice. Based on our experiments, ALT-MAS outperforms baselines on any class.

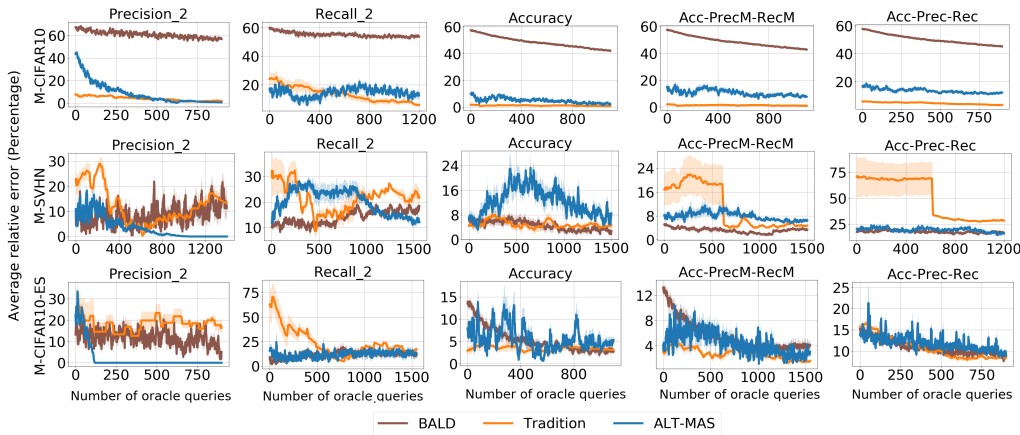

Figure 2: Average relative error of the estimated metrics, for each combination of model-under-test (*M-CIFAR10*, *M-SVHN*, & *M-CIFAR10-ES*) and metric set. Curves are smoothed over a sliding window of 10 oracle queries (iterations). Plotting mean and standard error over 3 repetitions. Better methods are those that converge faster to zero (Best seen in color).

**Tradition** performs very well on *M-MNIST* but performs badly on *M-FashionMNIST*. Based on our observation, **Tradition** performs well on *M-MNIST* because this model has very high accuracy (99.1%), which means its outputs and the true labels are very similar. Thus, by randomly picking data points, acquire their labels, and predict all the metric values by the mathematical formula on the current labelled set will provide the metric estimations to be approximately 100%, which are similar to the true values of the metrics. However, for other models-under-test with different accuracies, this technique will not work. Now, we compare the performance between **ALT-MAS** and **BALD**. For most of the combinations of model-under-test and metric set, our method **ALT-MAS** outperforms **BALD** by a high margin, especially on per-class metric and *M-MNIST*. This is understandable as i) *M-MNIST* is a good model and thus the BNN trained with **ALT-MAS** has much higher accuracy compared to that with **BALD**, and, ii) the optimal sampling strategy to achieve accurate estimations for per-class metrics is very different to the optimal sampling strategy to obtain a BNN with high accuracy. This clearly demonstrates the superiority of our BNN training and our optimal sampling methodologies. In particular, we can see that the BNN benefits the information from the model-under-test, and, the optimal sampling strategy to achieve accurate metric estimation should be the one that tailors to the characteristics of the models-under-test and the metrics.

## 4.2 CIFAR10 DATASET

We now consider the CIFAR10 test dataset (Krizhevsky, 2009). We use the same strategy as in Section 4.1 to generate three different models-under-test: *M-CIFAR10* (trained on train CIFAR10) with high accuracy, *M-CIFAR10-ES* (trained on train CIFAR10 with early stopping) with average accuracy, and *M-SVHN* (trained on SVHN - a different dataset) with low accuracy. We use the same 5 sets of metrics as in Section 4.1. The BNN we use in this case is the standard LeNet model (Lecun et al., 1998) with dropout applied before the relu layers. More details about the experiment setup are in Section C.2 of the appendix. In Figure 2, we report the average relative errors of the estimated metrics of several combinations of model-under-test and metric set. Similar to the observations on MNIST dataset, the method **Tradition** is not consistent, i.e. it performs well on the good model-under-test, but performs badly on other models-under-test. **BALD** also performs well on some scenarios but does not perform well when the model-under-test is good or when the metric is per-class metric. In contrast, **ALT-MAS** performs well on all of the scenarios.

## 4.3 THE QUALITY OF THE BNNS

In Section C.3 of the appendix, we investigate the performance of the BNNs in the active testing methods (**ALT-MAS** & **BALD**). In particular, we compare the classification accuracies of the BNNs in **BALD** and those in **ALT-MAS** (with our augmented data training strategy and optimal sampling strategy). First, we can see that, for most of the models-under-test and metric sets, the BNNs does not

need to have high accuracy for the metric estimation to be accurate. Second, due to our augmented data training strategy, the prediction accuracies of the BNNs improve significantly, especially when the models-under-test are good. Lastly, due to our proposed sampling strategy, the BNN only needs to learn the labels of the data points contributing significantly to the metric estimation in order to achieve accurate metric estimation. More discussions are in Section C.3 of the appendix.

## 5 RELATED WORK

**Trustworthy Machine Learning**. This line of works aims to assess the confidence of deep neural networks in making predictions on test data points (Platt, 1999; Zadrozny & Elkan, 2002; Niculescu-Mizil & Caruana, 2005; Guo et al., 2017; Gal & Ghahramani, 2016; Lakshminarayanan et al., 2017; Hendrycks & Gimpel, 2017; Jiang et al., 2018; Corbière et al., 2019). For example, (Gal & Ghahramani, 2016) and Lakshminarayanan et al. (2017) estimate the uncertainty of deep neural networks via Bayesian methods so as to return a distribution over the predictions. The work in Hendrycks & Gimpel (2017) uses the softmax probability of the network to detect if a test data point is misclassified or out-of-distribution. A new trust score to understand whether a classifier's prediction for a test data point can be trusted or not is proposed in Jiang et al. (2018). Most recently, Corbière et al. (2019) assess the confidence of a model by proposing a new target criterion for model confidence based on the True Class Probability. These methods rely on the training data and/or internal architecture of the network to generate the model confidence score. Our method, in contrast, assumes an already trained and black-box model; our goal is to estimate the performance of the model on various metrics on a new test dataset.

**Machine Learning Testing.** A comprehensive review of machine learning testing methods can be found in (Zhang et al., 2019). Some recent notable works include (Tian et al., 2018; Sun et al., 2018; Zhou et al., 2019). All of these works are based on white-box testing, i.e. the testing framework uses information about the internal structure of the model. Our work is black-box testing, i.e. our method assesses the performance of the model based solely on its outputs. The closest related work to ours are (Sawade et al., 2010; Schelter et al., 2020; Gopakumar et al., 2018) - these methods, their limitations, and how we overcome them are discussed in Section 1.

**Active Learning.** These methods aim to train a machine learning model in a data-efficient way by selecting the most informative data points for which labels should be acquired (Settles, 2010; Blundell et al., 2015; Gal et al., 2017). These AL methods aim to train a model to predict the labels of new data points accurately whilst our method aims to train a model to estimate a specified metric set accurately. We have shown experimentally that for the active testing framework, our proposed method outperforms existing AL methods.

## 6 CONCLUSION

We propose a novel approach to efficiently evaluate the performance of black-box machine learning models. The core idea is to efficiently estimate important metrics of the model being tested based on Bayesian neural network. We develop a novel method for training the BNN to achieve accurate metric estimations. We also devise a novel entropy-based sampling strategy to sample a data point such that the proposed framework can accurately estimate the metrics of interest simultaneously using a minimal number of labelled data. Experimental results show that our proposed approach works efficiently for estimating multiple metrics using diverse models and datasets.

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

## A APPENDIX

## B ALGORITHM DETAILS

### B.1 BACKGROUND

**Bayesian Neural Networks** Bayesian neural networks (BNNs) are special neural networks that maintain a distribution over its parameters (MacKay, 1992; Neal, 1995). In practice, performing exact inference in BNNs is generally intractable, thus, there has been a line of works aiming to perform approximate inference. In this work, we use the MC-dropout method (Gal & Ghahramani, 2016) to perform inference as it is both both scalable and theoretically guaranteed - equivalent to performing approximate variational inference to find a distribution in a tractable family that minimizes the Kullback-Leibler divergence to the true model posterior.

**Bayesian Active Learning by Disagreement (BALD)** BALD is a sampling methodology in Bayesian active learning, which samples the data point that maximizes the mutual information between the model prediction and the model posterior (Houlsby et al., 2011; Gal et al., 2017). To be more specific, let us denote $\mathcal{D}_l^t = \{\mathcal{X}_l^t, \mathcal{Y}_{\mathcal{X}_l^t}\}$ as the labelled set obtained after iteration $t$ and $\mathcal{B}_\omega^t$ as the BNN trained on $\mathcal{D}_l^t$ (with parameter $\omega$), then the acquisition function of BALD is defined as,

$$\mathbb{I}[y; \omega | x, \mathcal{D}_l^t] = \mathbb{H}[y | x, \mathcal{D}_l^t] - \mathbb{E}_{\omega \sim p(\omega | \mathcal{D}_l^t)}[\mathbb{H}[y | x, \omega]], \tag{10}$$

where $y$ is predicted label for the data point $x$. The data point that maximizes BALD is the data point for which the model has many possible predictions, i.e. the posterior draws have disagreement.

### B.2 DETAILS OF THE PROPOSED ACQUISITION FUNCTION

Our proposed acquisition function is defined as the mutual information (Houlsby et al., 2011; Gal et al., 2017) between $\tilde{Q}_k(x)$ and the BNN parameters $\omega$, i.e.,

$$\mathbb{I}[\tilde{Q}_k(x); \omega | x, \mathcal{D}_l^t] = \mathbb{H}[\tilde{Q}_k(x) | x, \mathcal{D}_l^t] - \mathbb{E}_{\omega \sim p(\omega | \mathcal{D}_l^t)}[\mathbb{H}[\tilde{Q}_k(x) | x, \omega]]. \tag{11}$$

This mutual information can be approximated using the MC-dropout variational distribution $q_\theta(\omega | \mathcal{D}_l^t)$. Next, we show how to approximate each tearm on the right hand side of Eq. (11).

**Computing $\mathbb{H}[\tilde{Q}_k(x)|x, \mathcal{D}_l^t]$**   For each data point $x$, $\hat{y}_x$ is a discrete random variable with $C$ distinct values, so $\tilde{Q}_k(x)$ is also a discrete random variable with at most $C$ distinct values. Therefore, $\mathbb{H}[\tilde{Q}_k(x)|x, \mathcal{D}_l^t]$ can be computed as,

$$\mathbb{H}[\tilde{Q}_k(x)|x, \mathcal{D}_l^t] = -\sum\nolimits_{q \in \mathcal{Q}} p(\tilde{Q}_k(x) = q|x, \mathcal{D}_l^t) \log p(\tilde{Q}_k(x) = q|x, \mathcal{D}_l^t), \tag{12}$$

where $\mathcal{Q}$ consists of all the possible values of $\tilde{Q}_k(x)$ when $\hat{y}_x \in \{1, ..., C\}$. By using the union bound, $\mathbb{H}[\tilde{Q}_k(x)|x, \mathcal{D}_l^t]$ can then be expressed as,

$$\mathbb{H}[\tilde{Q}_k(x)|x, \mathcal{D}_l^t] = -\sum\nolimits_{q \in \mathcal{Q}} \Big(\sum\nolimits_{h \in \tilde{Q}_k^{-1}(q)} p(\hat{y}_x = h|x, \mathcal{D}_l^t)\Big) \log \Big(\sum\nolimits_{h \in \tilde{Q}_k^{-1}(q)} p(\hat{y}_x = h|x, \mathcal{D}_l^t)\Big),$$

where $\tilde{Q}_k^{-1}(q)$ is the inverse function that maps the value of $\tilde{Q}_k(x)$ to $\hat{y}_x$. Given $M$ stochastic forward passes $\{\hat{\omega}_j\}$ from the MC-dropout posterior distribution $q_\theta(\omega|\mathcal{D}_l^t)$, $\mathbb{H}[\tilde{Q}_k(x)|x, \mathcal{D}_l^t]$ can finally be approximated as,

$$\begin{aligned}
\mathbb{H}[\tilde{Q}_k(x)|x, \mathcal{D}_l^t] \approx -\sum\nolimits_{q \in \mathcal{Q}} \Big(\Big(\sum\nolimits_{h \in \tilde{Q}_k^{-1}(q)} (\sum\nolimits_{j=1}^M p(\hat{y}_x = h|x, \hat{\omega}_j))/M\Big) \\
\times \log \Big(\sum\nolimits_{h \in \tilde{Q}_k^{-1}(q)} (\sum\nolimits_{j=1}^M p(\hat{y}_x = h|x, \hat{\omega}_j))/M\Big)\Big),
\end{aligned} \tag{13}$$

where $\tilde{Q}_k(x)$ can be approximated as $\tilde{Q}_k(x) \approx \mathbb{E}_{\omega \sim q_\theta(\omega|\mathcal{D}_l^t)}[Q_k(\mathcal{A}_\mathcal{X}, [\mathcal{Y}_{\mathcal{X}_l^t}, \hat{\mathcal{Y}}_{\mathcal{X}_{ul}^t \setminus x, \omega}, \hat{y}_x])] \approx (\sum_{j=1}^M Q_k(\mathcal{Y}_\mathcal{A}, [\mathcal{Y}_l^t, \hat{\mathcal{Y}}_{\mathcal{X}_{ul}^t \setminus x, \hat{\omega}_j}, \hat{y}_x]))/M$, with $\hat{\mathcal{Y}}_{\mathcal{X}_{ul_t}^t \setminus x, \hat{\omega}_j}$ denoting the predicted labels for $\mathcal{X}_{ul}^t \setminus x$ given the parameter $\hat{\omega}_j$.

**Computing $\mathbb{E}_{\omega \sim p(\omega|\mathcal{D}_l^t)}[\mathbb{H}[\tilde{Q}_k(x)|x, \omega]]$**   Similar as in the above paragraph, given $M$ stochastic forward passes $\{\hat{\omega}_j\}_{j=1}^M$ from the MC-dropout variational distribution $q_\theta(\omega|\mathcal{D}_l^t)$, $\mathbb{E}_{\omega \sim p(\omega|\mathcal{D}_l^t)}[\mathbb{H}[\tilde{Q}_k(x)|x, \omega]]$ can be approximated as,

$$\begin{aligned}
\mathbb{E}_{\omega \sim p(\omega|\mathcal{D}_l^t)}[\mathbb{H}[\tilde{Q}_k(x)|x, \omega]] &\approx \frac{1}{M} \sum\nolimits_{j=1}^M \mathbb{H}[\tilde{Q}_k(x)|x, \hat{\omega}_j] \\
&\approx -\frac{1}{M} \sum\nolimits_{j=1}^M \Big(\sum\nolimits_{q \in \mathcal{Q}} p(\tilde{Q}_k(x) = q|x, \hat{\omega}_j) \log p(\tilde{Q}_k(x) = q|x, \hat{\omega}_j)\Big) \\
&\approx -\frac{1}{M} \sum\nolimits_{j=1}^M \Big(\sum\nolimits_{q \in \mathcal{Q}} \Big(\sum\nolimits_{h \in \tilde{Q}_k^{-1}(q)} p(\hat{y}_x = h|x, \hat{\omega}_j)\Big) \\
&\qquad\qquad\qquad \times \log \Big(\sum\nolimits_{h \in \tilde{Q}_k^{-1}(q)} p(\hat{y}_x = h|x, \hat{\omega}_j)\Big)\Big),
\end{aligned} \tag{14}$$

where $\mathcal{Q}$ consists of all the possible values of $\tilde{Q}_k(x)$ when $\hat{y}_x \in \{1, ..., C\}$, $\tilde{Q}_k^{-1}(q)$ is the inverse function that maps the value of $\tilde{Q}_k(x)$ to $\hat{y}_x$, and $\tilde{Q}_k(x)$ can be approximated as $\tilde{Q}_k(x) \approx \mathbb{E}_{\omega \sim q_\theta(\omega|\mathcal{D}_l^t)}[Q_k(\mathcal{A}_\mathcal{X}, [\mathcal{Y}_{\mathcal{X}_l^t}, \hat{\mathcal{Y}}_{\mathcal{X}_{ul}^t \setminus x, \omega}, \hat{y}_x])] \approx (\sum_{j=1}^M Q_k(\mathcal{Y}_\mathcal{A}, [\mathcal{Y}_l^t, \hat{\mathcal{Y}}_{\mathcal{X}_{ul}^t \setminus x, \hat{\omega}_j}, \hat{y}_x]))/M$, with $\hat{\mathcal{Y}}_{\mathcal{X}_{ul_t}^t \setminus x, \hat{\omega}_j}$ denoting the predicted labels for $\mathcal{X}_{ul}^t \setminus x$ given the parameter $\hat{\omega}_j$.

## C   EXPERIMENT DETAILS

### C.1   MNIST DATASET

**Models-under-test**   The three models-under-test we use to evaluate our proposed method are:

(i) *M-MNIST*: A CNN trained on the train MNIST dataset (LeCun et al., 2010). We implement the model in tensorflow but use the official network architecture published on keras.com[4] as there is no official source code on tensorflow.org. The accuracy of this model on the test MNIST dataset is 99.06%;

---

[4]https://keras.io/examples/mnist_cnn/

(ii) *M-MNIST-ES*: A CNN trained on the train MNIST dataset. The model architecture is same as M-MNIST, but we use early stopping to decrease the model performance, i.e. we set smaller epochs and higher batch size. The accuracy of this model on the test MNIST dataset is 70.67%;

(iii) *M-FashionMNIST*: A CNN trained on the train FashionMNIST dataset (i.e. a completely different dataset compared to MNIST) (Xiao et al., 2017). The model is implemented using the official code published on tensorflow.org.[5] The accuracy of this model on the test MNIST dataset is 12.39%.

**Metric sets** The metric sets we use to evaluate our proposed method are:

(i) Three sets of metrics consisting of only one metric in each set: precision of class 2, recall of class 2, and overall accuracy. These are common metrics used to evaluate performance of classifiers, and these metrics cover both per-class metrics and overall metrics.

(ii) Two sets of metrics consisting of multiple metrics. We use two sets: one set consisting of 3 metrics (accuracy, macro-precision, macro-recall), and one set consisting of 21 metrics (accuracy, precision and recall of each class).

**The BNN and the binary classifier architecture** The BNN and the binary classifier have the same architecture. The architecture is an MLP with two layers, 256 neurons/layer, and with dropout applied in each layer. The number of MC-dropout samples are 50. The initial labelled set $\mathcal{D}_l^0$ has 100 data points randomly sampled from the test dataset. We tuned 3 hyper-parameters: learning rate, epochs and dropout rate using Bayesian optimization (Snoek et al., 2012). We also reinitialize the BNN after each iteration as in (Gal et al., 2017).

## C.2 CIFAR10 DATASET

**Models-under-test** The three models-under-test we use to evaluate our proposed method are:

(i) *M-CIFAR10*: A CNN trained on the train CIFAR10 dataset (Krizhevsky, 2009). We implement the model in tensorflow but use the official network architecture published on keras.com[6] as there is no official source code on tensorflow.org. The accuracy of this model on the test CIFAR10 dataset is 77.43%.

(ii) *M-CIFAR10-ES*: A CNN trained on the train CIFAR10 dataset. The model architecture is same as M-CIFAR10, but we use early stopping to decrease the model performance, i.e. we set smaller epochs and higher batch size. The accuracy of this model on the test CIFAR10 dataset is 40.39%.

(iii) *M-SVHN*: A CNN trained on the train SVHN dataset (i.e. a completely different dataset compared to CIFAR10) (Netzer et al., 2011). The model architecture is same as *M-CIFAR10*. The accuracy of this model on the test CIFAR10 dataset is 9.3%.

**Metric sets** The metric sets we use are same as the metrics set used for the MNIST dataset (as described in Section C.1).

**The BNN and the binary classifier architecture** The BNN and the binary classifier have the same architecture. The architecture is the standard LeNet model (Lecun et al., 1998) with dropout applied before the relu layers. The number of MC-dropout samples are 50. The initial labelled set $\mathcal{D}_l^0$ has 500 data points randomly sampled from the test dataset. We tuned 3 hyper-parameters: learning rate, epochs and dropout rate using Bayesian optimization (Snoek et al., 2012). We also reinitialize the BNN after each iteration as in (Gal et al., 2017).

## C.3 THE QUALITY OF THE BNNS

In Figures 3 and 4, we plot the prediction accuracies of the BNNs in **BALD** and **ALT-MAS** for all combinations of models-under-test and metric sets on the MNIST and CIFAR10 dataset. Below,

---

[5]https://www.tensorflow.org/tutorials/keras/classification
[6]https://keras.io/examples/cifar10_cnn/

we evaluate how complex the BNN needs to be for the metric estimation to be accurate and how effective our augmented training strategy and our proposed sampling methodology are.

Theoretically, we can see that for the BNN to be useful, it does not need to have high classification accuracy; it only needs to accurately predict the data points that contribute to the metric estimation. Being incorrect at other data points does not affect the predicted metric values. See example:

| True label | Model-under-test prediction | Surrogate model prediction |
|---|---|---|
| 0 | 5 | Any prediction except 5 (e.g. 1, 2, 3, ...) will give accurate metric estimation |
| 5 | 5 | Correct prediction matters for some metrics (e.g. accuracy, precision/recall of class 5), but not for other metrics (e.g. precision/recall of other classes) |

In summary, the metric estimation accuracy depends on the model-under-test and the metric set. This is clearly demonstrated on the performance of **BALD**. With **BALD**, we use the predicted labels by the BNNs trained by **BALD** to estimate the metric values. For MNIST, the BNNs trained with **BALD** have accuracies ranging from $70 - 90\%$, but for the models-under-test *M-FashionMNIST* and *M-MNIST-ES* (average & bad models), the metric estimation accuracies range from $90 - 100\%$ - that is much higher than the BNNs' accuracies. Similarly, for CIFAR10, the BNNs' accuracies range from $40 - 60\%$, but for the models-under-test *M-SVHN* and *M-CIFAR10-ES*, the metric estimation accuracies range from $90 - 100\%$. For the good models-under-test *M-MNIST* and *M-CIFAR10* then BALD requires high BNN accuracy in order to achieve good metric estimations.

For our proposed method **ALT-MAS**, with the models *M-FashionMNIST*, *M-MNIST-ES*, *M-SVHN* and *M-CIFAR10-ES* and the overall metrics (accuracy), the observations are similar to **BALD**. That is, the BNNs accuracies are much lower to the metric estimation accuracies. The better thing is that, for the per-class metrics (precision and recall of class 2, the BNNs accuracies do not need to be high (as in the overall metrics) in order to achieve high metric estimation accuracy. In particular, the BNNs accuracies by **ALT-MAS** are much lower than the BNNs by **BALD**, but the metric estimations by **ALT-MAS** are much higher than by **BALD**. This proves the motivation of our sampling approach, that is, the BNN only needs to accurately predict the data points that contribute to the metric estimation. For the good models-under-test *M-MNIST* and *M-CIFAR10*, due to our augmented training strategy, the BNN accuracies improve significantly, and thus, leads to the metric estimations to be much more accurate compared to **BALD**.

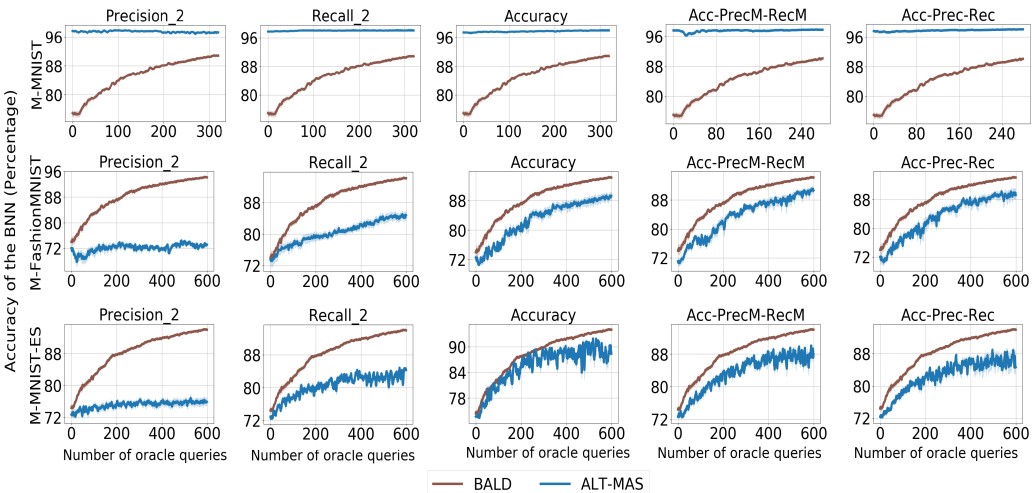

Figure 3: The accuracy of the BNN, for each combination of model-under-test (*M-MNIST*, *M-FashionMNIST*, & *M-MNIST-ES*) and metric set. Curves are smoothed over a sliding window of 5 oracle queries (iterations). Plotting mean and standard error over 3 repetitions (Best seen in color).

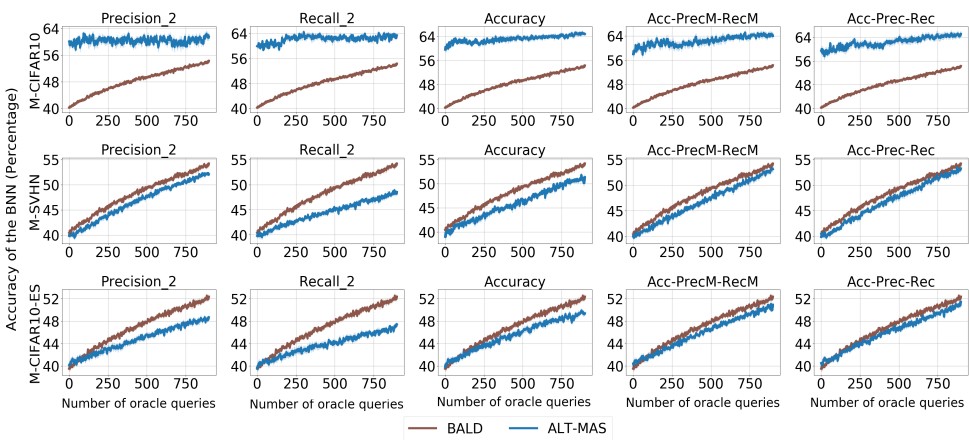

Figure 4: The accuracy of the BNN, for each combination of model-under-test (*M-CIFAR10*, *M-SVHN*, & *M-CIFAR10-ES*) and metric set. Curves are smoothed over a sliding window of 5 oracle queries (iterations). Plotting mean and standard error over 3 repetitions (Best seen in color).

