# OpenReview forum: "ALT-MAS: A Data-Efficient Framework for Active Testing of Machine Learning Algorithms"
_ICLR.cc/2021/Conference — Reject_

### Official Review · AnonReviewer1 · 2020-10-19
**I recommend to reject the paper, mostly because the experimental results do not support the claim that the proposed approach performs much better than traditional evaluation or prior works. Furthermore, appropriateness of the experimental setup and novelty of the BNN training are unclear and proper discussion of related works is missing.**

**Rating:** 3
**Confidence:** 4

**Review:**

#### Summary:
The paper proposes an active testing approach that actively selects test instances to estimate the performance of a (black box) machine learning model. The key idea is to train a Bayesian Neural Networks (BNN) with a small amount of labeled test data and evaluate how well the model-under-test agrees with the BNN on samples for which the BNN has a high confidence. More instances to be labeled by an oracle are selected with active learning, i.e. select the data point that minimizes the uncertainty of the metric prediction.

#### Recommendation:
I recommend to reject the paper, mostly because the experimental results do not support the claim that the proposed approach performs much better than traditional evaluation or prior works. Furthermore, appropriateness of the experimental setup and novelty of the BNN training are unclear and proper discussion of related works is missing.

#### Strong Points:
- Active testing is an important research direction since more and more pre-trained models are applied in practice without being fine-tuned.
- The approach is clearly described. The paper is well-written and easy to understand.

#### Weak Points:
- The evaluation does not support the claim that ALT-MAS performs substantially better than traditional evaluation or prior works. Out of 30 plots, the proposed method performs best in approx. 3 cases, ties for the first place in approx. 8 cases, and is inferior (wrt. the traditional eval and BALD) in the remaining ~19 cases.
- The title suggests that the paper addresses machine learning in general. However, only a simple deep neural network for MNIST and CIFAR10 is used in the evaluation. Hence, it remains unclear if the approach can successfully be applied to other machine learning methods and datasets. In general, it would be a strong point of this paper to include many different machine learning models since it is agnostic regarding the learning framework used.
- I disagree with the general statement 'Better methods are those that converge faster to zero'. It depends a lot on the budget available for labeling new instances and the desired value (and confidence) of the target metrics. A clearly better method would have a lower error for any amount of labeling budged.
- Active testing is appealing when it can be assumed that the test data has a different distribution than the training data. However, most of the experiments use same train/test distribution. Hence, the approach is not evaluated in an appropriate setting.
- The paper claims to contribute 'novel approach to train the BNN so as to accurately estimate the metrics of interest'. However, it uses Monte Carlo dropout. It is unclear which part of the training is novel.
- The main body of the paper does not contain a section that positions the work well in the available literature.

#### Questions:
1. Do I understand correctly that the 'Traditional' evaluation uses as many samples for evaluation as ALT-MAS and BALD? I am referring to the phrase 'the traditional method where the metrics are computed using their mathematical formula with all the labeled data *up to the current iteration*, and the labeled data is picked randomly from the whole test dataset'.
2. In Section 4.2, you write that 'ALT-MAS performs well on all of the scenarios'. In Figure 2, first row, I'd say that ALT-MAS is inferior than the traditional evaluation in 4 out of 5 cases and ties for the first place for Recall_2. How do you come to the conclusion that ALT-MAS performs well on all of the scenarios?

---

> ### Author Response · Authors · 2020-11-19
> **Answers to AnonReviewer 1 - Part 1**
>
> We thank the reviewer for the detailed comments. We address the reviewer's questions below, and we are happy to continue discussing any of these points or answer follow-up questions.
>
> 1. The evaluation does not support the claim that ALT-MAS performs substantially better than traditional evaluation or prior works.
>
> We’d like to emphasize that our proposed method ALT-MAS only performs worse than other baselines (either Traditional or BALD) in several cases; and in these cases, ALT-MAS only slightly performs worse (please refer to the y-axis value of each plot) - Please refer to our answer in Question 3 for how to define better method. Besides, as discussed in the paper, one of the key advantages of ALT-MAS is that it is the most consistent method. It performs consistently well in all cases while other methods do not perform consistently well in all cases, i.e. the average relative errors of ALT-MAS are always within a reasonable range whilst other methods can perform very badly in some cases. In particular,
> + The method ‘Traditional’ performs very badly when the models-under-test are bad models (the 2nd row of Figures 1 & 2). The average relative error of ‘Traditional’ is extremely large in few cases (can go up to 1e7 %). On the other hand, ALT-MAS performs well in all cases, its average relative errors are always within a reasonable range.
> + The method ‘BALD’ performs very badly when the model-under-test is good (the 1st row of Figures 1 & 2). The average relative error of ‘BALD’ is large in all these cases (can go up to 60%). In contrast, the average relative error of ALT-MAS is always within a reasonable range in all cases.
>
> 2.	The title suggests that the paper addresses machine learning in general. However, only a simple deep neural network for MNIST and CIFAR10 is used in the evaluation.
>
> We’d like to clarify that we evaluated the methods using 6 different models-under-test (3 models-under-test for each dataset) – See Sections C.1 and C.2 in the Appendix for descriptions of these models-under-test. For each model-under-test, we evaluate 5 different sets of metrics. This means we have evaluated in total 30 different settings – which is extremely extensive.
>
> 3.	I disagree with the general statement 'Better methods are those that converge faster to zero'.
>
> We agree that a clearly better method would have a lower error for any amount of labelling budget, however, in practice, the situations are more complex. For example, there are methods that perform well (smaller errors) at the beginning when the number of labelled data is small, but do not perform well when the number of labelled data is larger or goes to infinity. For data-efficient methods, it is common to evaluate the performance of the methods asymptotically, i.e. to analyse the performance of the methods when the number of labelled data increases to infinity. This evaluation criterion has been commonly used in many areas such as active learning, Bayesian optimization. This is the reason why we stated: “Better methods are those that converge faster to zero”.
>
> 4.	Active testing is appealing when it can be assumed that the test data has a different distribution than the training data. However, most of the experiments use same train/test distribution.
>
> We’d like to emphasize that in the experiments, we did evaluate the scenario when the test data has a different distribution compared to the training data. Specifically, for each dataset, we evaluate all the methods with a model-under-test that were trained on a completely different dataset (with 5 different sets of metrics) – See Sections 4.1, 4.2 in the main paper and Sections C.1 and C.2 in the Appendix. This evaluation is already 1/3 of all our experiments.
> It is also worth noting that the purpose of our framework is to assess the performance any machine learning model-under-test, not just model-under-test when training data distribution is different with test data distribution. Thus, as discussed in Sections 4.1 and 4.2, we evaluate the proposed framework using different types of models-under-test: good model (training dataset is same as testing dataset and perfect training process), average model (training dataset is same as testing dataset and average training process), bad model (training dataset is different with testing dataset).

---

> > ### Author Response · Authors · 2020-11-19
> > **Answers to AnonReviewer 1 - Part 2**
> >
> > 5.	The paper claims to contribute 'novel approach to train the BNN so as to accurately estimate the metrics of interest'. However, it uses Monte Carlo dropout. It is unclear which part of the training is novel.
> >
> > The novelty of the BNN training process is the augmented training data methodology. Section 3.1 is devoted to describing this novel augmented training data methodology. Specifically, we proposed to train a binary classifier that aims to predict the data points in the test dataset for which the model-under-test agrees with the ground-truth. We then use the predictions from this binary classifier to construct an augmented labelled set and combine this augmented labelled set with the original labelled data to train the BNN. It has been shown in the experiments that because of this novel augmented training data methodology, the metric estimations of our proposed method are more accurate – See the discussion in Sections 4.1 and 4.2.
> >
> > 6.	The main body of the paper does not contain a section that positions the work well in the available literature.
> >
> > Due to the space limit, we put the section ‘Related work’ in the Appendix. We now have revised the paper to add the Related work section to the main paper. Please refer to Section 5 for a discussion on the related literature.
> >
> > Other questions:
> >
> > 7.	Do I understand correctly that the 'Traditional' evaluation uses as many samples for evaluation as ALT-MAS and BALD?
> >
> > Yes, the ‘Traditional’ method uses as many samples for evaluation as ALT-MAS and BALD. The differences between ‘Traditional’ and ‘ALT-MAS’+’BALD’ are that, for ‘Traditional’: 1) the samples are drawn randomly from the test dataset, and 2) the estimated metrics are computed using the mathematical formula of the metric functions.
> >
> > 8.	In Section 4.2, you write that 'ALT-MAS performs well on all of the scenarios'. In Figure 2, first row, I'd say that ALT-MAS is inferior than the traditional evaluation in 4 out of 5 cases and ties for the first place for Recall_2. How do you come to the conclusion that ALT-MAS performs well on all of the scenarios?
> >
> > In the first row, Traditional does perform better than ALT-MAS, however, the difference is not much significant (the difference in average relative error is less than 10%). On the other hand, in 2nd row, when ‘Traditional’ performs badly, the difference between ‘Traditional’ and ‘ALT-MAS’ is very significant (the difference in average relative error can go up to 50%). This is the reason we write ‘ALT-MAS performs well on all of the scenarios’, that is, ALT-MAS consistently gives good metric estimation in all scenarios whilst ‘Traditional’ can perform very badly in some scenarios.

---

### Official Review · AnonReviewer3 · 2020-10-28
**Nice problem setting and good performance.**

**Rating:** 6
**Confidence:** 4

**Review:**

Authors proposed ALT-MAS, a data-efficient testing framework that can accurately estimate the performance of a machine learning model, a novel approach to train the BNN to accurately estimate the metrics of interest, and a novel sampling methodology to estimate the metrics of interest efficiently. The performances of proposed methods are demonstrated through the empirical effectiveness of our proposed machine learning testing framework on various models-under-test for a wide range of metrics and different datasets.

The problem setting addressed by the authors is one of the hot topics. The experimental results on the two data sets show good performance on the proposed method.

Experimental results using MNIST and CIFAR10 show that the proposed method can consistently provide better accuracy than the conventional method.

Although the proposed method is based on early stopping, we could not be sure from the paper whether the proposed method can consistently reproduce the same performance on other tasks.

---

> ### Author Response · Authors · 2020-11-19
> **Answers to AnonReviewer 3**
>
> We thank the reviewer for the encouraging feedback. We address the reviewer's questions below, and we are happy to continue discussing any of these points or answer follow-up questions.
>
> 1. The problem setting addressed by the authors is one of the hot topics. The experimental results on the two data sets show good performance on the proposed method.
>
> We thank the reviewer for the kind words.
>
> 2. We could not be sure from the paper whether the proposed method can consistently reproduce the same performance on other tasks.
>
> In our experiments, we have evaluated our proposed framework using 6 different models-under-test (3 models-under-test for each dataset) – See Sections C.1 and C.2 in the Appendix for the descriptions of these models-under-test. For each model-under-test, we evaluate 5 different sets of metrics. This means we have evaluated in total 30 different settings – which is extremely extensive. With this extensive evaluation, we believe our proposed method will perform well in many tasks.

---

### Official Review · AnonReviewer4 · 2020-10-28
**Review of ALT-MAS**

**Rating:** 4
**Confidence:** 5

**Review:**

Summary: The authors have proposed using an active learning approach to estimate evaluation metrics for a given model. The approach learns a sampling function that decides which observations need to be labeled, which are then fed to a Bayesian neural network (BNN) that aims to estimate the distribution Y|X. The authors select which observations to sample by maximizing the mutual information between the model evaluation metric and the BNN parameters.

Pros:
+ Active testing of models is a difficult problem in high-dimensions. The proposed problem is highly relevant.

Major concerns:

1. The authors have primarily focused on comparing against a traditional sampling method that simply takes IID samples and a deep-learning-based active learning algorithm for learning the distribution of Y|X. Both of these comparators are very weak. The former is not doing any active learning. The latter is trying to do a much harder task of learning Y|X rather than estimating the evaluation metrics. I think a more reasonable comparison method is the active testing approach from Sawade 2010 that estimates the distribution p(Y|X;\theta) using a neural network.

2. There seem to be a number of misconceptions about prior work. The authors argue that previous approaches are specific to particular evaluation metrics and cannot scale to high-dimensional data. However, the active risk estimation approach taken in Sawade 2010 is quite general and can be used to estimate the expected value of any function l that maps the predicted target and the true target to a real-valued number, which is the same type of functions considered in this paper. It is also easy to extend the approach to multi-variate evaluation metrics by, say, replacing the squared error with the squared L2 norm. Finally, you can use any estimator of Y|X in their framework, so one could use a neural network to do active risk estimation for high-dimensional data.

3. The authors do not give a justification for why they used mutual information to prioritize which observations to label. Is there a way to show that it minimizes the estimation error of the evaluation metrics? And similarly, why should one try to prioritize which observations to label using the sum of mutual informations?

4. I am confused by the y-axes in Figures 1 and 2. What is average relative error? The average relative error is very large in certain cases. In this case, do all the methods have unacceptably large errors?

5. When estimating evaluation metrics for classifiers, it is important to characterize the theoretical guarantees of the estimates. Do we know if the method is consistent or (asymptotically) unbiased? Are there confidence intervals associated with this approach? In addition to some discussion of the theoretical properties, please evaluate the bias of the method in simulations (whereas the simulations currently only show relative error).

---

> ### Author Response · Authors · 2020-11-19
> **Answers to AnonReviewer 4**
>
> We thank the reviewer for the thoughtful comments. We address the reviewer's questions below, and we are happy to continue discussing any of these points or answer follow-up questions.
>
> 1. Comparision with the method in [Sawade 2010]?
>
> A comparison with Sawade 2010 is not possible as this work only provides tractable formulas for two specific risk functions (l_{0/1} norm for classification problems and l_2 norm for regression problems) – See Derivations 1 & 2 in Section 3.1 of Sawade 2010. For classification problems, the l_{0/1} norm corresponds to the accuracy metric – See Eqs. (1) & (2) in Section 2 of Sawade 2010. This means that for classification problems, the method in Sawade 2010 can only be used to sample data so as to estimate the accuracy metric. In contrast, our proposed method can estimate any metrics (e.g. accuracy, per-class metric, macro-precision, macro-recall).
>
> 2. There seem to be a number of misconceptions about prior work?
>
> The work in Sawade 2010 did derive a general formula to actively sample data points for any risk function that maps the predicted target and the true target to a real-valued number (Theorem 1 in Section 3.1), however, it is unclear if this formula is tractable for every function. To be specific, this general formula depends on the unknown ground truth of the data in the test dataset, the unknown metric estimation, and on the unknown true conditional p(Y|X). In Sawade 2010, the paper only suggests two tractable derivations of this general formula when the risk function is l_{0/1} norm and l_2 norm (Derivations 1 & 2 in Section 3.1). For classification problems, the risk function l_{0/1} norm corresponds to the case when the metric needs to be estimated is the accuracy metric. This is the reason why we argued in our paper that previous approaches are specific to some particular evaluation metrics. We have revised our paper to make this point clearer – See the 2nd paragraph in the Introduction Section.
>
> 3. The use of mutual information to prioritize which observations to label?
>
> In deep Bayesian active learning (AL), mutual information is one of the most common criteria used to select the data point to label [Gal et al, 2017]. Specifically, the data point that maximises the mutual information can be (heuristically) considered as the data point with the highest prediction uncertainty [Gal et al, 2017]. Selecting this data point will maximally improve the prediction accuracy of the trained model in the AL problem. In our work, we construct the acquisition function (Eq. 5) by deriving a new random variable that represents the metric estimation and combining with the mutual information criterion. The data point that maximises this acquisition function can be (heuristically) considered as the data point that causes the highest uncertainty in the metric estimation. By labelling this data point, we maximally reduce the uncertainty in the metric estimation, and thus, maximally improve the accuracy of the metric estimation.
>
> The same logic can be applied in the case of multiple metrics. That is, we aim to select the data point that maximises the sum of mutual information so as to maximally improve the accuracy of all the metric estimations. We have revised our paper to clarify this point clearer.
>
> 4.  What is average relative error?
>
> The average relative error is the average relative error of the metric estimations. Assume we need to estimate K metrics with the true values being m_1, …, m_K, and if we denote the estimated metrics are \hat{m}_1, …, \hat{m}_K, then the average relative error is computed as 1/K (|\hat{m}_1 - m_1| / m_1 + … + (|\hat{m}_K - m_K| / m_K). This error is large in certain cases as in some cases the true metric value is relatively small, and thus, it makes the average relative error to be large. We use the average relative error to compare between methods because for a metric set, the metric values are in different ranges, and thus, relative error needs to be used.
>
> 5. Characterize the theoretical guarantees of the estimates?
>
> Theoretically, when the number of data points in the test dataset is finite, we have shown that the estimation obtained by our method is unbiased (Remark 3.3). When the number of data points in the test dataset is infinite, we currently don’t have the theoretical analysis. It’s worth noting that for deep Bayesian AL, deriving the asymptotically theoretical analysis is a very challenging task and most of the current work on deep Bayesian AL do not have the asymptotically theoretical analysis. In our future work, we will continue to explore this analysis.
>
> Empirically, we can see that the average relative error converges to 0 when the number of labelled data increases to infinity (Figures 1 & 2). From the formula of the average relative error, it can be directly inferred that when the average relative error converges to 0, the absolute error of each metric estimation also converges to 0.

---

### Official Review · AnonReviewer2 · 2020-10-29
**This paper aims to use active learning to obtain labels for points that allow for the most efficient way of estimating metrics of interest describing the performance of a classifier. The idea is that this would be simpler and more efficient than the typical active learning strategy of trying to learn a model to actually predict what the labels would be. The paper demonstrates this through some experiments.**

**Rating:** 8
**Confidence:** 4

**Review:**

The paper is quite clear and is well-motivated by a practical situation of having a classifier that is a black box and that is evaluated by various metrics that one would like to assess as efficiently as possible. The paper demonstrates that the simpler task of using active learning to identify examples for labeling that would most reduce the uncertainty in metrics of interest is more effective than the more general task of using active learning to learn to predict the labels that the original classifier would give and using those results to calculate the metrics of interest.

The only cons that I see are that the paper lacks some obvious explorations that I think would be quite valuable and informative:
1. What is the variation in the sequences of points chosen for labeling depending on the metric that is being calculated?
2. Calculating the ROC curve does not require classifier internal structure. It only requires some continuous output representing class membership rather than just a discrete indication.
3. Remark 3.2: Simplicity seems an insufficient reason to have the binary classifier and Bayesian Neural Network have the same structure. This should be explored and at least a summary of performances given for variations on this.
4. To sample the point that is best for multiple metrics, equation (9) is used, which calculates the sum of the mutual informations between the BNN parameters and the metrics. Is it obvious that using the sum is the best way? I wonder if using something different like the point that most increases the minimum mutual information may work better.
5. In algorithm 1, step 6, note that $\mathcal{C}_{\eta}^{t-1}$ is being trained on $\mathcal{D}_l^{t-1}$.
6. In algorithm 1, step 8, $\mathcal{S}_l^{t-1}$ is undefined. Do you mean to refer to $\mathcal{S}_l$ from step 7?
7. In figure 1, the average metrics seem to vary quite significantly with the number of oracle queries. Since you are measuring the average, I expected the decrease to be relatively smooth. How do you account for the significant variation?
8. The Gal and Ghahramani paper is listed twice in the bibliography.

---

> ### Author Response · Authors · 2020-11-19
> **Answers to AnonReviewer 2**
>
> We thank the reviewer for the encouraging feedback.  We address the reviewer's questions below, and we are happy to continue discussing any of these points or answer follow-up questions.
>
> 1. What is the variation in the sequences of points chosen for labelling depending on the metric that is being calculated?
>
> The variation in the sequences of points chosen for labelling is caused by the variation in the initial labelled set D^0_l. In each experiment, we initialize with a different labelled set D^0_l (randomly selected from the whole test dataset), and thus, the sequences of points selected for labelling are different in each experiment. This variation is the variation in the average relative errors in Figures 2 & 3.
>
> 2. Calculating the ROC curve does not require classifier internal structure. It only requires some continuous output representing class membership rather than just a discrete indication.
>
> We agree with this point. We have corrected this in the revised version of the paper.
>
> 3. Remark 3.2: Simplicity seems an insufficient reason to have the binary classifier and Bayesian Neural Network have the same structure. This should be explored and at least a summary of performances given for variations on this.
>
> As both the binary classifier and the BNN have the same type of inputs, thus, a simple choice is to use the same architecture for these two models. Note that, even when the two models share the same architecture, their hyper-parameters (e.g. number of training epochs, drop-out rate, learning rate) are very different. In particular, we employed Bayesian optimization to tune the hyper-parameters for each model given the labelled data – See Sections C.1 and C.2 for more details. Should time and computing resources permit, we will make our best efforts to obtain a summary of the performances when the binary classifier architecture varies.
>
> 4. To sample the point that is best for multiple metrics, equation (9) is used, which calculates the sum of the mutual information between the BNN parameters and the metrics. Is it obvious that using the sum is the best way? I wonder if using something different like the point that most increases the minimum mutual information may work better.
>
> As the reviewer points out, when we have the mutual information corresponding to each metric, combining them using the sum is perhaps the easiest, and most practical way. This is also our motivation when constructing the acquisition function for a set of metrics by summing the mutual information. As shown in our experiments, this idea works well in all scenarios. And thank you for the suggestion of selecting the data point that most increases the minimum mutual information, we will try this idea in our future work.
>
> 5. In algorithm 1, step 6, note that Cηt−1 is being trained on Dlt−1.
> 6. In algorithm 1, step 8, Slt−1 is undefined. Do you mean to refer to Sl from step 7?
>
> Yes, in step 6, C_η^{t−1} is being trained on D_l^{t−1}. And in step 8, we meant to refer to S_l from step 7. There is a typo in step 7, it should be S_l^{t-1} (not S_l). We have fixed these two points in the revised version of the paper.
>
> 7. In figure 1, the average metrics seem to vary quite significantly with the number of oracle queries. Since you are measuring the average, I expected the decrease to be relatively smooth. How do you account for the significant variation?
>
> In deep Bayesian active learning, since the BNN is retrained in each iteration, it is common that there is variation in the prediction accuracy (see the performance of BALD in Figures 1 & 2). In our proposed method, besides the BNN, we also trained a binary classifier for the augmented data step, thus, the variation is larger compared to BALD.
>
> 8. The Gal and Ghahramani paper is listed twice in the bibliography.
>
> Thanks for spotting this! We have fixed this in the revised version of the paper.

---

### Author Response · Authors · 2020-11-19
**General remark**

We thank all the reviewers for their thoughtful comments. We have noticed that many of the reviews are quite positive, and we really appreciate these comments. We also noticed some negative comments which we believe came from our insufficient explanation on the prior work and experimental settings. We have explained these concerns in the Author Response and we have also updated the submission following the reviews. We hope that the reviewers can re-evaluate our paper after reading our answers. We are very happy to continue discussing any of the concerns or answer follow-up questions.

---

### Decision · Program_Chairs · 2021-01-07
**Final Decision**

**Decision:**

Reject

**Comment:**

The paper addresses a well-motivated problem of evaluating the accuracy of a black-box classifier A(x) using actively selected set of labeled examples.  They predict two additional classifiers --- one to predict if an example will be corrected by A(x) and the second a Bayesian NN to assign a distribution over likely labels of unlabeled examples.  The final accuracy estimate is obtained   from the labeled data and the probabilistic labels on the unlabeled data.
Reviewers rightly pointed out that the paper only compares with conventional active learning methods, and skips comparison with methods specifically designed for active evaluation.  A list is attached below.  Also, the overall technical contribution seems limited in terms of both empirical accuracy gains it obtains and novelty of ideas exposed.

Related papers:
A lazy man's approach to benchmarking: Semisupervised classifier evaluation and recalibration P Welinder, M Welling, P Perona - Proceedings of the IEEE …, 2013 - cv-foundation.org

Active evaluation of classifiers on large datasets N Katariya, A Iyer, S Sarawagi - 2012

Adaptive Stratified Sampling for Precision-Recall Estimation. A Sabharwal, Y Xue - UAI, 2018

Online Active Model Selection for Pre-trained Classifiers Mohammad Reza Karimi, Nezihe Merve Gürel, Bojan Karlaš, Johannes Rausch, Ce Zhang, Andreas Krause

Towards Efficient Evaluation of Risk via Herding Z Xu, T Yu, S Sra